# Usefulness of Vaccine Adverse Event Reporting System for Machine-Learning Based Vaccine Research: A Case Study for COVID-19 Vaccines

**DOI:** 10.3390/ijms23158235

**Published:** 2022-07-26

**Authors:** James Flora, Wasiq Khan, Jennifer Jin, Daniel Jin, Abir Hussain, Khalil Dajani, Bilal Khan

**Affiliations:** 1Department of Computer Science and Engineering, California State University San Bernardino, 5500 University Parkway, San Bernardino, CA 92407, USA; 006981423@coyote.csusb.edu (J.F.); jennifer.jin@csusb.edu (J.J.); khalil.dajani@csusb.edu (K.D.); 2School of Computer Science and Mathematics, Liverpool John Moores University, Liverpool L3 3AF, UK; w.khan@ljmu.ac.uk; 3Division of Vascular & Interventional Radiology, Department of Radiology, Loma Linda University Medical Center, Loma Linda, CA 92354, USA; djin@llu.edu; 4Department of Electrical Engineering, University of Sharjah, Sharjah P.O. Box 27272, United Arab Emirates; abir.hussain@sharjah.ac.ae; 5Institute of the Environment and Sustainability, University of California Los Angeles, Los Angeles, CA 90095, USA

**Keywords:** COVID-19, VAERS, adverse events, vaccine development, association rule mining, self-organizing maps, hierarchical clustering, bipartite graphs, vaccine analysis workflow

## Abstract

Usefulness of Vaccine-Adverse Event-Reporting System (VAERS) data and protocols required for statistical analyses were pinpointed with a set of recommendations for the application of machine learning modeling or exploratory analyses on VAERS data with a case study of COVID-19 vaccines (Pfizer-BioNTech, Moderna, Janssen). A total of 262,454 duplicate reports (29%) from 905,976 reports were identified, which were merged into a total of 643,522 distinct reports. A customized online survey was also conducted providing 211 reports. A total of 20 highest reported adverse events were first identified. Differences in results after applying various machine learning algorithms (association rule mining, self-organizing maps, hierarchical clustering, bipartite graphs) on VAERS data were noticed. Moderna reports showed *injection-site*-related AEs of higher frequencies by 15.2%, consistent with the online survey (12% higher reporting rate for *pain in the muscle* for Moderna compared to Pfizer-BioNTech). AEs {*headache*, *pyrexia*, *fatigue*, *chills*, *pain*, *dizziness*} constituted >50% of the total reports. *Chest pain* in male children reports was 295% higher than in female children reports. *Penicillin* and *sulfa* were of the highest frequencies (22%, and 19%, respectively). Analysis of uncleaned VAERS data demonstrated major differences from the above (7% variations). Spelling/grammatical mistakes in allergies were discovered (e.g., ~14% reports with incorrect spellings for *penicillin*).

## 1. Introduction

VAERS, an online passive reporting system, co-sponsored by the US Center for Disease Control and Prevention (CDC) and the Food and Drug Administration (FDA), and the agencies of US Health and Health Services (HHS) are specifically geared towards assessing the safety of newly developed vaccines along with other priorities that include: (i) the detection of new, unusual, or rare vaccine adverse events, (ii) the monitoring of the increase in known events, (iii) the identification of potential risk factors for particular types of adverse events (AEs), (iv) the determination of possible reporting clusters, (v) the recognition of persistent safe-use problems, and (vi) the provision of national safety monitoring to public health emergencies, such as a large-scale pandemic influenza vaccination program [1,2,3]. Due to its spontaneous reporting nature, VAERS data is not recommended for discerning the cause of AEs from the vaccine after an AE is reported. Although the availability and utilization of high-quality vaccine data for decision support and vaccine safety is critical, public reports prior to a vaccine authorization by VAERS can be useful in determining AEs and in providing valuable insights for a streamlined vaccine manufacturing and policy development.

VAERS datasets have been used in various studies for recommendations and proactive strategies for regulatory bodies (CDC and FDA) [4,5,6,7,8,9,10,11,12,13,14,15,16,17,18,19,20,21]. To date, only limited studies have comprehensively focused on the protocols to be followed when VAERS datasets are used for statistical analyses (Appendix A). For example, a study compiled VAERS reports on Guillain-Barre Syndrome (GBS) in regard to influenza vaccines and identified correlations between the AE and the syndrome, including the attributes age and gender [14]. Two distinct datasets were utilized (i.e., 80,059 US (VAERS FLU3, 1990–2017) reports and 13,550 European reports (all FLU vaccination, 2003–2016)) to develop a logistic regression model for predicting 83 different AEs with prediction accuracies of 77.5% and 75.5% (area under the curve (AUC) measures) for VAERS and European FLU vaccine datasets, respectively. Patient age (as quantized into the ranges of {0.5–17, 18–49, 50–64, and 65+} years) and gender were considered as model attributes. Syndrome to AE correlation was carried using Chi-squared test which demonstrated nine AEs (*pyrexia*, *chills*, *nausea*, *pruritus*, *rash*, *urticaria*, *injection site pain*, *injection site swelling*, and *injection site erythema*) to be negatively associated with GBS while 13 other AEs (*muscle spasms*, *hypertension*, *dysphagia*, *hyperglycemia*, *diabetes mellitus*, *dysuria*, *depression*, *apnea*, *fecal incontinence*, *constipation*, *urinary incontinence*, *dysuria*, *urinary tract infection*, and *urinary retention*) were positively associated with GBS but with low prevalence (<1%). The study acknowledged that VAERS data are screened by the CDC for the removal of duplicates.

A study emphasized that the identification of duplicate pairs in VAERS for the application of data-mining algorithms on VAERS data that, without robustly handling duplicate cases, can have deleterious effects on quantitative analyses leading to spurious conclusions on vaccine safety [22]. A probabilistic approach was developed to link duplicate pairs allowing a systematic approach of deduplicating the VAERS database using the structured field data as well as non-structured textual data of AEs via event-based text-mining approach. Another useful analysis of the validity of VAERS reports via expert judgement was carried out that demonstrated the lower likeliness of an AE being associated with a vaccine [19]. A total of 100 VAERS reports of the AE following immunization (AEFI) were analyzed where 83% achieved majority agreement over the results of the causality assessment, while 17% of the reports were considered for further discussion by the expert panel. From the 100 reports, 3%, 20%, and 20% of the AEFI were identified as being definitely, probably, and possibly related to the vaccine, respectively, while 53% of the AEFIs were classified as unlikely or unrelated to the vaccine.

Data provenance methods and preprocessing techniques based on only a passive reporting system require careful attention when carrying out data-driven exploratory analysis and applying statistical approaches on VAERS data in order to avoid misleading/incorrect conclusions. The factors contributing to the robust and accurate analyses of such data include the handling of: (i) duplicate records, (ii) missing values (submitting incomplete VAERS forms), (iii) limited-form fields (up to 5 symptoms in one report) leading to duplicates, (iv) spelling/grammatical mistakes via robust and appropriate text mining approaches, (v) outliers and data standardization/normalization, (vi) data heterogeneity, and (vi) the binning of continuous variables (such as age) into groups to avoid bias when applying probabilistic/frequentist approaches. Accordingly, the present study proposes the aforementioned data provenance and preprocessing techniques for robust statistical analyses with the help of a case study for COVID-19 vaccine data collected from VAERS. Dynamic trends in unstructured temporal COVID-19 vaccine data from VAERS were analyzed via self-organizing maps (SOMs), association rule mining (ARM), and hierarchical clustering (HC) techniques in order to provide a detailed data-driven evaluation of multi-AE associations and complex patterns. Reports from VAERS and a qualitative online survey were incorporated to: (i) identify the frequently reported AEs after COVID-19 vaccines, (ii) assess their correlations with respect to various demographics (age groups, gender, and allergies), and (iii) provide a baseline decision support for predictive capability when deidentified data become available from regulatory agencies as well as the vaccine producers. Such analysis can be useful for determining the proportion of reports involving specific AEs and a vaccine can be compared to the proportion of reports involving the same AEs and other vaccines [2].

## 2. Results

Figure 1a,b shows the relative frequencies of the 20 most-reported AEs for all age groups per three vaccine manufacturers and children of ages up to (and inclusive of) 15 years old, respectively. AEs for each vaccine manufacturer were significantly consistent. There were 13 AEs {*arthralgia*, *asthenia*, *chills*, *dizziness*, *dyspnoea*, *fatigue*, *headache*, *injection site pain*, *myalgia*, *nausea*, *pain*, *pain in extremity*, *pyrexia*} that were common for all three vaccine manufacturers. Survey data also reported {*headache*, *aches*, *chills*, *pain in muscle*, *dizziness*, *nausea*, *vomiting*, and *rash*} to be the most commonly reported AEs (Table 1). *Rash* was replaced by *injection site pain* for children’s data (Figure 1b) when duplicates and spelling mistakes were corrected in the VAERS reports.

Interestingly, four *injection-site*-related AEs {*injection site—*(*erythema*, *pruritus*, *swelling*, *warmth*)} were among the top 20 AEs for Moderna (*p*-value < 2.2 × 10^−16^ for Moderna vs. {Pfizer-BioNTech, Janssen} with respect to the top 20 AEs including *injection-site*-related AEs). Survey data also showed 51% of the samples for Moderna with *pain in muscle* as opposed to only 39% samples for Pfizer-BioNTech reporting *pain in muscle* (Table 1). This may be due to the fact that Moderna uses a 100-microgram dose as opposed to the 30-microgram used by Pfizer-BioNTech, causing increased reactogenicity [23]. Additionally, although the etiology of delayed large local reactions due to Moderna is unclear, a delayed-type hypersensitivity reaction to the excipient polyethylene glycol can be a potential etiology [24]. The above visual exploration without duplicate-row removal (Appendix A) showed relative frequencies of the above 20 AEs to be lower by up to 7% (Appendix A) than the frequencies observed in the cleaned data (Figure 1a). Similarly, the AEs for children showed differences of up to 4% (Figure 1b and Appendix A). 

The subset {*dizziness*, *pyrexia*, *headache*, *nausea*, *vomiting*, *fatigue*, *dyspnoea*, *pain*, *pain in extremity*, *chills*, *rash*} was common among adults (including children (Figure 1a)) and children (Figure 1b). None of the injection-site-related AEs {*injection site*—(*pain*, *erythema*, *swelling*, *warmth*)} were among highly reported AEs in children’s reports. Additionally, {*arthralgia*, *asthenia*, *myalgia*, *pruritus*, *erythema*} only appeared in the 20 most reported AEs for adults where {*chest pain*, *syncope*, *loss of consciousness*, *pallor*, *hyperhidrosis*, *urticaria*, *fall*, *unresponsive to stimuli*, *myocarditis*} were reported only among children. The above differences may arise due to the Pfizer-BioNTech dose for children being only 10 micrograms compared to 30 micrograms for adults.

As given in the heatmap in Table 2, although in a different order based on their percentage, all 20 highest-reported AEs for both children’s genders were the same. An important pattern in children’s VAERS reports was found to have *chest pain* reported to be 3 times higher in male reports than in female reports (chi-squared test *p*-value: 5.74 × 10^−62^). Based on the number of occurrences, *vomiting* was ranked as the top effect for female as opposed to the 2nd for male children (*p*-value: 0.56 indicating no significant correlation of vomiting with gender). It is noted, however, that for the VAERS dataset with duplicates, *headache* appeared as the top-ranked effect for female (Appendix A) as opposed to 5th-ranked for male children (*p*-value: 0.61). Additionally, *injection site pain* ranked a level higher for female (6th) compared to male children (7th), with *p*-value: 0.59 (i.e., no significant correlation of *injection site pain* with gender). Other correlation tests with *p*-values are {*Dizziness:* 3.8 × 10^−14^, *Pyrexia:* 8.21 × 10^−6^, *Fatigue:* 4 × 10^−3^, *Nausea:* 4 × 10^−4^, *Pain in Extremity:* 0.77, *Rash:* 0.88, *Pain:* 0.075, *Chest Pain:* 5.74 × 10^−62^}. It is also noted that the ratio of female VAERS COVID-19 reports is higher than male reports, which is consistent with other VAERS vaccine ratios (e.g., flu vaccine for 2021 had the number of reports as female: 5222, and male: 2375).

When grouped into clusters via an unsupervised HC approach, male children and young adults (i.e., age groups of 18 inclusive and under) were clustered in one group (i.e., Cluster III with blue dendrogram), as shown in Figure 2. For male children, {*dizziness*, *headache*, *pyrexia*} were grouped in the same cluster (Cluster II) with {*nausea*, *vomiting*} to be in the adjacent cluster (Cluster III), consistent with the grouping provided in Table 2. Furthermore, {*fatigue*, *chills*, *pain*} for male children were clustered in Cluster I. Interestingly, the HC approach demonstrated tolerance in grouping datasets with and without duplicates, as no difference in Figure 2 and Appendix A was observed. Overall, VAERS reports for male participants in Clusters I and II (*fatigue*, *chills*, *pain*, *dizziness*, *headache*, *pyrexia*} were to be of the highest percentage, as confirmed in Table 2. Consequently, due to {*dizziness*, *headache*, *nausea*, *pyrexia*} being reported more commonly between the age groups of 12–15 and 16–18 for female, they were grouped in the same cluster as shown in Figure 2, while 5–11 grouped in adjacent cluster. Consistent with Table 2, *injection-site*-related AEs in female and male children were grouped in clusters I and IV with lower-reporting percentages in Figure 2 and Appendix A, respectively.

It is noted that, despite comprehensive data preprocessing steps, reports submitted through VAERS have not undergone data-quality assurance/control strategies, thus posing challenges for the verification of the analysis. To overcome the challenge of the uncertainty and reliability of the VAERS reports and confirm the AE similarities, an exploration of the online survey data was also conducted (Figure 3). As illustrated in Table 1 and Figure 3, from a set of 11 AEs compiled from 211 participants, {*headache*, *chills*, *dizziness*, *nausea*, *itchy skin*/*rash*, *vomiting*} also appeared in the 20 most reported AEs in the VAERS reports.

### 2.1. Associations of the Most Commonly Reported AEs via ARM and SOM

The interrelationships of AEs from VAERS reports were analyzed via ARM and SOM with respect to two major age groups [<16, ≥16]. Assessment of the interrelationships of AEs for children revealed 16 non-redundant association rules (ARs) (Table 3). From a subset of one-to-one rules, the existence of *Hyperhidrosis* or *flushing* was shown to imply the existence of *dizziness* with lift-over 3 (*R*_2,10_). *Chest pain* was found to be prominent with dependency over the subset {*Electrocardiogram ST segment elevation*, *Chest X-ray normal*, *Echocardiogram normal*, *Myocarditis*, *Electrocardiogram normal*, *C-reactive protein increased*, *Troponin increased*} with a lift value of >8 (*R*_3,5,7–9,11,13_). Additionally, it was also noticed that *hyperhidrosis* was associated with *flushing* with a high lift value of 18.8 (*R*_9_). Although fatigue appeared among the top 6 AEs for children based on its individual frequency, its correlation with any other AE could not qualify it for the top 14 ARs (Table 3 and Figure 1).

The ARM employs a frequentist approach to calculate the *Support*, *Confidence*, and *Lift* (Appendix A), for which duplicate reports can pose a significant challenge. Therefore, a new report or reports with spelling/grammar mistakes can impact the generality and specificity of the ARs, impacting such analysis with duplicates present in the VAERS data. As illustrated in Appendix A, the ARs for children and each vaccine producer indicated significant differences from those identified when duplicates were removed (Table 3 and Table 4). For example, *R*_6_ (*Echocardiogram normal → Troponin increased*) for the non-redundant ARs for children (Table 3) demonstrated that the highest *lift* value of 16.6 was initially not identified as a non-redundant AR in Appendix A. Additionally, seven rules (*R*_3,5,7–9,11,13_) reported *chest pain* in the consequent cleaned VAERS data for children (Table 3) whereas none of the ARs in Appendix A reported *chest pain* in the consequent VAERS data with duplicates. ARs (*R*_4,10,12,14_) when verified via SOM in Figure 4 demonstrate the relationships of {{*Unresponsive to stimuli* → *Syncope*}, {*Hyperhidrosis* → *Dizziness*}, {*Chills* → *Pyrexia*}, {*Headache*, *pain* → *Pyrexia*}}. However, SOM may also suffer from misleading correlations from uncleaned VAERS data due to the iterative nature of 2D-map refinement (Appendix A and Figure 4).

Analysis of the ARs for the AEs of all age groups was also conducted for the three vaccine types (Table 4). In the set of ARs for Pfizer-BioNTech, *headache* appeared in the consequent of 10 ARs, with {*chills*, *myalgia*, *pyrexia*, *pain*, *fatigue*, *nausea*} in antecedents with count values > 3000 (*R*_5,6,8,10–16_). Although the above distributions appeared to be dispersed without demonstrating a discernible pattern (Figure 4), the overall distributions showed similarities in the SOM component planes. However, with duplicates present, only two ARs (*R*_19,20_) had *headache* in the consequent (Appendix A), due to the fact that the entries for *headache* were distributed with duplicates, increasing the frequency with which *headache* appeared. Another observation (Table 4) showed 7 out of 25 ARs for Moderna listed *injection-site*-related effects (e.g., *injection site* {*pruritus*, *pain*, *induration*, *warmth*, *swelling*, *erythema*}) in either the antecedent or the consequent with *Injection site swelling* → *injection site erythema* (*R*_5_) having the second highest count of 13,561. Additionally, the distributions for {*injection site* (*erythema*, *pain*, *swelling*), *pain in extremity*} were also interrogated via SOM (Appendix A) to validate the existence of correlations among these AEs as indicated by the rules (*R*_1,2,4,5,7,8,15,16_) in Table 4. The similarity between AEs as represented by the 2D SOM is indicative of the coexistence of their correlations (i.e., the existence of a base AE implies the existence of another AE as given by the distributions on a 2D map).

ARs for Janssen (Table 4) showed that 6 of 14 ARs reported *headache* in the consequent, with {*fatigue*, *pain*, *pyrexia*, *chills*, *myalgia*, *nausea*} appearing in the antecedent (Appendix A). This is in contrast with Pfizer-BioNTech and Moderna, where the AR with highest count was *chills* → *pyrexia*, *pyrexia* → *headache* had the highest count of 6574 (*R*_10_). An interesting AR *R*_2_ indicated a noteworthy observation {*decreased appetite* → *fatigue*}, with a 609 count value for Janssen. The AR *R*_10_ was also demonstrated with the help of SOM (Appendix A) showing similarity for *pyrexia* and *headache*, despite the lack of indication of definitive clusters in the SOM.

### 2.2. Interrelations of Vaccine AEs via Bipartite Graphs

The interrelationships between the 20 most commonly reported AEs and the three vaccines were also interrogated via bipartite graphs [25,26,27] (Figure 5, Appendix A and Appendix A). As shown in Figure 5, *headache* was most often reported for all 3 vaccines with a relative existence of 11%. The *injection-site*-related AEs {*injection site* (*erythema*, *pain*, *swelling*)} are of a higher relative percentage for Moderna (5%, 6%, and 4%) compared to those for Pfizer-BioNTech (1%, 4%, and 1%) and Janssen (1%, 3%, and 1%). The relationships of allergies with the 20 most-reported AEs in Figure 5e showed *penicillin* and *sulfa* to have the highest occurrences with 22% and 19% frequency, respectively. In the same figure, *penicillin* and *sulfa* appear to be uniformly distributed among all 20 AEs with *headache*, *fatigue*, and pyrexia having the highest percentages. Additionally, *gluten* from 3% of VAERS reports demonstrated a correlation with *fatigue* in 11% of data after cleaning and data pre-processing steps. Such a percentage suggests that the AR of *gluten* with *fatigue* may be supported with a higher level of confidence than the AR of *sulfa* with *fatigue*. Studies have reported that a significant percentage (31%) of patients with a self-reported *gluten* sensitivity had a lack of energy (third-highest symptom). Reports with non-coeliac *gluten* sensitivity also appear to correlate with {*depression*, *anxiety*, *headache*, *fatigues*, *reflux*, and *irritable bowel syndrome*} [25]. One study found that 82% of those newly diagnosed with coeliac disease complained of *fatigue*. Limited literature also indicates that *fatigue* can potentially be caused by *malnutrition*, induced by *intestinal damage* causing *malabsorption* of nutrients [26]. *Fatigue* can also be caused by *anemia*, which frequently appears in patients with coeliac disease [27].

It is noted that the VAERS data that included 5 distinct symptoms reported as 5 attributes in free-form text were of significant percentage with spelling mistakes. For example, *penicillin* was reported with various spelling variations such as {*penecellin*, *penecillin*, *penecilin*}, and sulfates was reported as {*sulfa*, *sulpha*, *sulfides*, *sulfite*, *sulfate*}. Another notable spelling mistake in the present analysis was the use of words “*vaccination site*” and “*injection site*” interchangeably such as *vaccination site* {*pain*, *mass*, *induration*, *swelling*, *warmth*, *inflammation*} and *injection site* {*pain*, *mass*, *induration*, *swelling*, *warmth*, *inflammation*}. The words “*vaccination site*” were replaced with “*injection site*” for consistency.

## 3. Discussion

The usefulness of the VAERS data for the statistical analysis of vaccines was illustrated with the help of a case study for COVID-19 vaccine data. It was emphasized that, due to the specific reporting format by VAERS online submission portal, its passive nature and access to the public can have an impact on any machine-learning (ML)/data-mining approach when careful data preprocessing approaches are omitted (i.e., removing/merging duplicates in VAERS, discretizing numeric attributes, handling missing values, and fixing spelling/grammar errors). With the help of these data provenance and preprocessing techniques, it is hoped that vaccine research and development can utilize and streamline the protocols when ML techniques are applied to VAERS data. The present study proposes a set of recommendations supported by the application of various ML algorithms that are critical to applying modeling approaches to or exploratory analyses of VAERS data. An online survey was also conducted, providing 211 distinct reports of the COVID-19 post-vaccination effects from participants in the US. Various useful data preprocessing/cleaning techniques were pinpointed, which should be considered to be part of VAERS.

It is noted that, although models of various types have been developed for different vaccine reports based on exploratory data analyses and the application of ML techniques on VAERS data [4,6,7,9,10,11,12,13,15,16,20,28], the model development for evolving VAERS data can be exposed to unseen situations that would neither be available for model training nor for validation. Despite the anticipated outcome from the ML perspective, the monitoring and testing strategies should be carefully implemented. Studies utilizing VAERS data for vaccine safety based on ML techniques require the following best practices.

### 3.1. Flexibility offor Model Features

Data and model-feature provenance strategies should be documented, including feature definitions, data ranges, meta-level requirements, and privacy controls. Structure of the developed ML model should be made flexible for new feature addition and updates to existing features.

### 3.2. Robust Model-Development Pipelines

Model development for vaccine AE identification and predictive capability should be reviewed, tested, and updated for the continuous refinement of existing workflows. Modularity in terms of model applicability on all or selected slices of data should be accomplished through a robust development pipeline, and model parameters should be tuned upon the availability of new data.

### 3.3. ML Model Verification

In order to enhance model applicability and reproducibility, validation (via unit, system, and integration testing) should be ensured before deployment into the production environment, or any policy or recommendation is proposed. Appropriate model maintenance and documentation strategies should be implemented, and transparency in terms of step-by-step debugging (on single data instances) should be demonstrated.

### 3.4. Model Stability and Efficiency

Model efficiency should be carefully evaluated via robust tests to ensure the reasonable use of computational resources in order to provide accurate predictions. Such tests can be based on model-training speed, use of RAM, and throughput in a real-time learning environment. Additionally, automation test cases can be developed to verify model prediction accuracy and stability (in terms of predictive accuracy) over time, as well as latency issues.

## 4. Materials and Methods

Analysis of the psychological and physical effects of COVID-19 vaccines along with the discovery of correlations among the most commonly reported AEs was conducted as per the workflow described in Figure 6. Vaccine data for Pfizer-BioNTech, Moderna, and Janssen were obtained via VAERS, which was accompanied by a primary dataset collected from an online survey comprising information on post-vaccine AEs and public perception of the COVID-19 vaccine. Online survey data were designed to fill data gaps in the absence of other closely monitored data repositories such as v-safe [29], whose data have not yet been made available for public and research communities. The overarching goal of the present study of VAERS and the online survey data was to pinpoint critical data provenance and management protocols for robust statistical analysis and predictive modeling of vaccines with a case study of COVID-19 vaccines. Particular steps to assess the efficacy of data-driven techniques applied on VAERS data were based on: (i) the exploration of the post-vaccination effects of COVID-19 vaccines on various age groups (particularly children under the age of 16), (ii) the determination of the frequencies of reported AEs after each dose of COVID-19 vaccines, (iii) the evaluation of the co-existence of common post-vaccine AEs via unsupervised ML approaches, and (iv) the assessment of potential relationships of pre-existing conditions (e.g., allergies) with the AEs. Active reporting via an online survey was also aimed for to further assess the impact of COVID-19 vaccination via the reported AEs, evaluate psychological perception of COVID-19 vaccination, and compare the VAERS reports with an active and systematically controlled system that incorporates quality data into COVID-19 vaccine domain knowledge.

### 4.1. Compilation, Preprocessing, and Exploration of VAERS Data

Two distinct datasets were compiled with 905,976 and 211 data samples from VAERS (filtered to prune rows for the three COVID-19 vaccines) and an online survey, respectively. The VAERS reports consisted of vaccine- and patient-related attributes that included vaccine identification (VAX type, VAX manufacturer, VAX lot, VAX does series, VAX route, VAX site, VAX name), free-form textual attributes {US state, gender, allergies, hospital, disability, current illness}, binary attributes {birth defects, prior visit, ER visit}, age (numeric), and vaccination date (date). VAERS reports that did not list any AEs were removed, reducing the dataset size to 892,213 reports. Data cleaning was then performed to merge duplicate reports and fix spelling/grammar mistakes, resulting in a total of 643,522 reports. The age attribute was discretized into 7 groups (5–11, 12–15, 16–18, 19–30, 31–50, 51–65, and 66+) for the purpose of identifying the age-to-AE correlation via bipartite plots (Section 3.3). Data statistics per manufacturer for each of the above age groups and genders are given in Table 5 and Appendix A, along with the numbers of categories for each attribute in both datasets from VAERS (original without removing duplicates (Appendix A) and after data preprocessing (Table 5)) and the online survey. A summary of the content of the datasets (without the removal of duplicate records) is also provided in Appendix A, which lists the number of data samples for each of the 20 most commonly reported AEs along with their percentage per manufacturer.

VAERS reports for children of age under 16, with a total of 12,489 VAERS samples, were also collected and analyzed separately in order to explore the commonality between the AEs with respect to different age groups. The goal of this analysis was to discover meaningful patterns (i.e., AEs) that appear collectively in children when compared to adults or differences as the age group progresses to an older population. Data from children’s reports were also cleaned where rows that reported any attribute (column) from {*age group*, *gender*, *symptom*, and *vaccine manufacturer*} as “unknown” were removed. Additionally, reports indicating “*product administered to patient of inappropriate age*” while reporting no AEs were also removed. Cleaned data after the removal of reports with “*product administered to patient of inappropriate age*” comprised of 9457 reports with distributions of 9142, 228, and 87 for Pfizer-BioNTech, Moderna, and Janssen, respectively (Table 5). The AEs submitted in children’s VAERS reports were also separated in the form of heatmaps (Table 2) with respect to the gender in order to identify gender similarities/dissimilarities with the help of cell colors based on the percentage of the corresponding AEs. The AEs for all genders in Table 2 were sorted based on the age group (Section 2 and Section 2.1) of 5–11 years old. A non-cleaned version of the VAERS reports (i.e., the reports with duplicates) is provided in Appendix A.

### 4.2. Exploratory Data Analysis of the COVID-19 Vaccines’ Effects

The initial exploratory analysis for VAERS data was conducted to determine the frequencies of AEs to support advanced analysis. The 20 most commonly reported AEs were first used to assess their associations, as shown in Table 1. Similar to Table 5 and Appendix A, statistics based on non-cleaned data are reported in Appendix A, demonstrating significant differences from Table 1, which could have a significant impact on the performance and robustness if a statistical approach is applied.

### 4.3. Correlation Analysis of AEs Based on Age Groups and Allergies

Unsupervised ML approaches utilizing ARM and SOMs (Appendix A) were applied on VAERS and survey data, where the endpoints were analyzed to explore the relationships among AEs and reported allergies. Unsupervised learning is useful for visual data exploration to find hidden data groups in order to better understand the correlation of the AEs with existing medical conditions without any predictions or testing the underlying hypotheses. ML approaches are also helpful for applying statistical approaches to cluster/group similar biological effects to enhance the applicability domain of the vaccines as well as recommend proactive strategies for vaccine safety. Furthermore, as new data become available, analyzing the relationships among post-COVID-19 vaccine AEs and other reported demographical characteristics via ML approaches can be helpful in designing improved versions of vaccines (e.g., COVID-19 pills) for COVID-19 vaccine safety. Mapping the relationships (i.e., associations) among the reported post-COVID-19 vaccine AEs via unsupervised ML techniques is particularly helpful in revealing useful patterns, streamlining COVID-19 vaccine safety standards and the development of robust models for proactive strategies and recommendations. Through these relationships, one can assess the co-occurrence of certain AEs and infer the reasons that the emergence of one or more AE may lead to other AE(s) that are correlated due to biological or other relevant reasons. The ARM of AEs was also accompanied by confirmatory cluster analysis approaches based on hierarchical clustering.

ARM has been applied in various disciplines [28,30,31,32,33,34,35,36,37]. Irrespective of the domain of interest, triggering of one or more AE can imply the triggering of other AEs, consistent with the crosstalk between various physical AEs and perceptional indicators. The ARM of the AEs after each vaccine dose can be used to identify many-to-many relationships and propose a data-driven hypothesis-generation technique. A detailed description of ARM can be found in the Appendix A. ARs in the present study were also validated with the help of the SOM analysis, demonstrating the VAERS data distribution on 2D maps. Cluster analysis via SOMs has been demonstrated to be useful for discovering relationships in complex multidimensional datasets in cross-disciplinary areas of research and development [38,39,40,41]. SOM clustering applies competitive learning, preserves topological structure of the input space, and transforms the output to a lower dimension (i.e., 2-D map of cells within SOM clusters). Further discussion on SOM can be found in the Appendix A. The utility of SOMs for data visualization and feature selection has also been demonstrated for exploratory data analyses [34,38,39,41,42,43,44,45,46,47]. For the analysis via ARM and SOM, open-source libraries were utilized, which are freely available online (R Studio arules—version 1.7-3 [48] (for ARM), kohonen version 3.0.11 [49] (for SOM analysis), hclust version 3.6.2 [50] (for HC), and Python stats.chisquare [51] (for statistical significance test)).

The interrelationships of the AEs with allergies and other personalized factors (age group and gender) were identified via bipartite graphs (Section 2.2 and Appendix A). Bipartite graphs established in the present study are useful for the exploratory analysis of potential allergies, age groups and genders that may be indicative of the occurrence of one or more common AEs. Moreover, bipartite graphs allow for the bidirectional exploration of COVID-19 vaccine data for detailed information about specific AEs and their causal (i.e., {allergy, age group, gender} → AE) or a diagnostic reasoning (i.e., AE → {allergy, age group, gender}). Graphical displays of correlations between reported AEs and allergies can help explore the frequencies of certain AEs, interrogate comparisons between them and their occurrences given certain pre-existing conditions, identify similarity/distribution among reports that demonstrated similar AEs, and assess potential causes of AEs given certain pre-existing conditions [47]. For example, it can be seen in Figure 5 that *Age group of* 31–50 years old has been reported to have the highest percentage (36%) among all of the 20 commonly reported AEs. Each bar in the bipartite graph is further split into sub-bars representing its distribution in terms of the available categories for each of the three variables {age group, gender, and allergies} across 20 AEs. The bars on the left side of the bipartite graphs list the 20 most commonly reported AEs. The bipartite graphs in the present study were created using the open-source JavaScript library from d3.js [52].

## Figures and Tables

**Figure 1 ijms-23-08235-f001:**
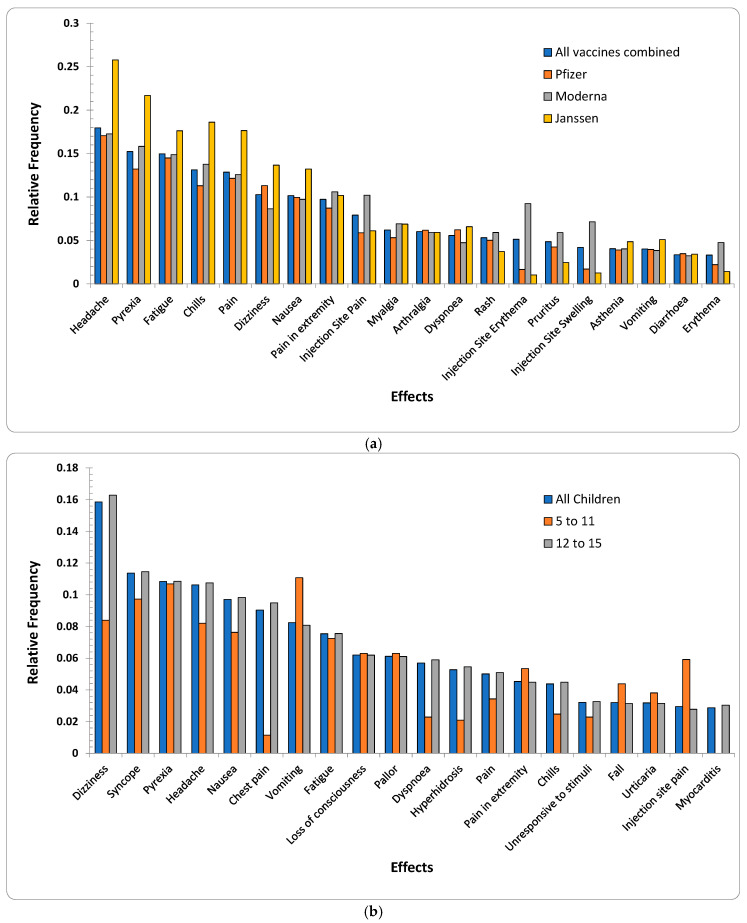
(**a**) Relative frequencies of the top 20 AEs appeared in VAERS reports for all age groups per the three vaccine producers (Pfizer-BioNTech, Moderna, and Janssen). (**b**) Relative frequencies of the top 20 AEs appeared in VAERS reports for children (discretized age groups of 5–11 years). The subset {*chest pain*, *Dyspnoea*, *hyperhidrosis*, and *myocarditis*} was among the lowest-reported AEs for age group (5–11 years) in comparison to the AEs reported for the group 12–15 and other 16 most commonly reported AEs.

**Figure 2 ijms-23-08235-f002:**
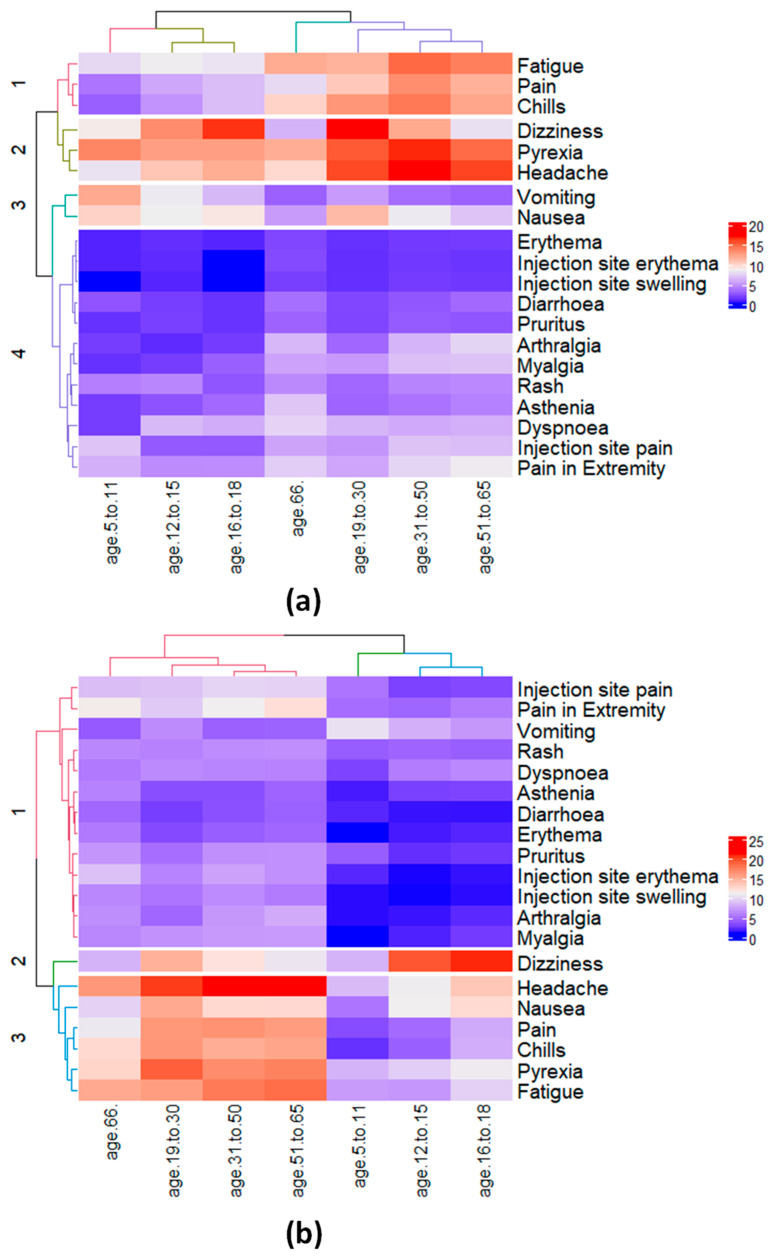
Hierarchical clustering of the 20 most commonly reported effects and 7 age groups for (**a**) male and (**b**) female participants. For male participants (**a**) the effects {*pyrexia*, *vomiting*, and *nausea*} and {*dizziness*, *pyrexia*} were reported to be the most commonly reported effects for the two children age groups 5–11 and 12–15, respectively. For female participants (**b**), the effects {*headache*, *pyrexia*, *nausea*, *vomiting*, and *dizziness*} were reported to be the most commonly reported effects for children age group 5–11 and 12–15, respectively, with the addition of *nausea* among the 3rd most-reported effect for age group 12–15.

**Figure 3 ijms-23-08235-f003:**
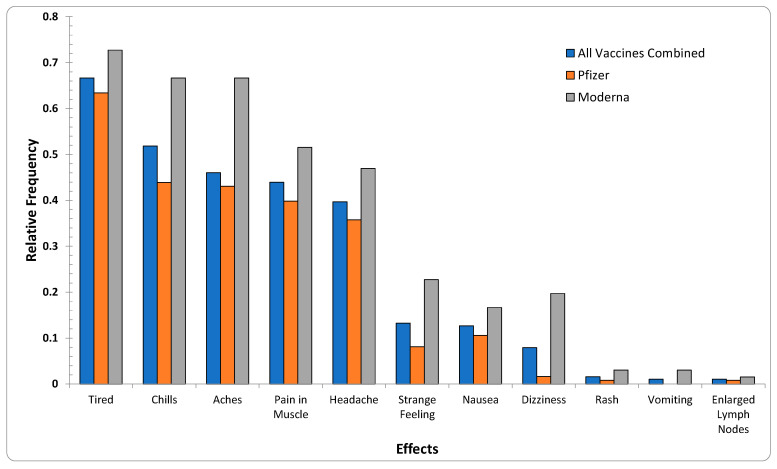
Relative frequencies of the 11 AEs appearing in survey data reports for all age groups per the two vaccine producers (Pfizer-BioNTech, Moderna). The subset {*headache*, *chills*, *dizziness*, *nausea*, *itchy skin*/*rash*, *vomiting*} was the same as 6 of the 20 most-reported effects in VAERS reports.

**Figure 4 ijms-23-08235-f004:**
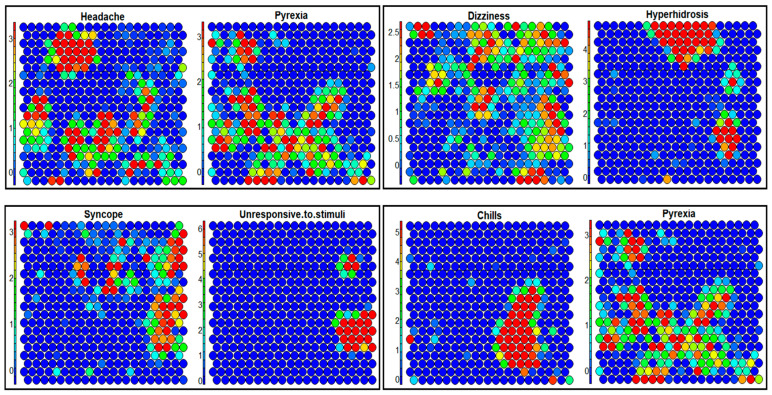
SOM analysis of the top 20 most reported AEs from the reports filtered for children (age 15 (inclusive) and under) for all vaccines. Association of AEs {*chills*, *dizziness*, *headache*, *hyperhidrosis*, *pyrexia*, *syncope*} in the form of 2D cluster similarities is demonstrated as shown in the rules *R*_4,10,12,14_ in Table 3.

**Figure 5 ijms-23-08235-f005:**
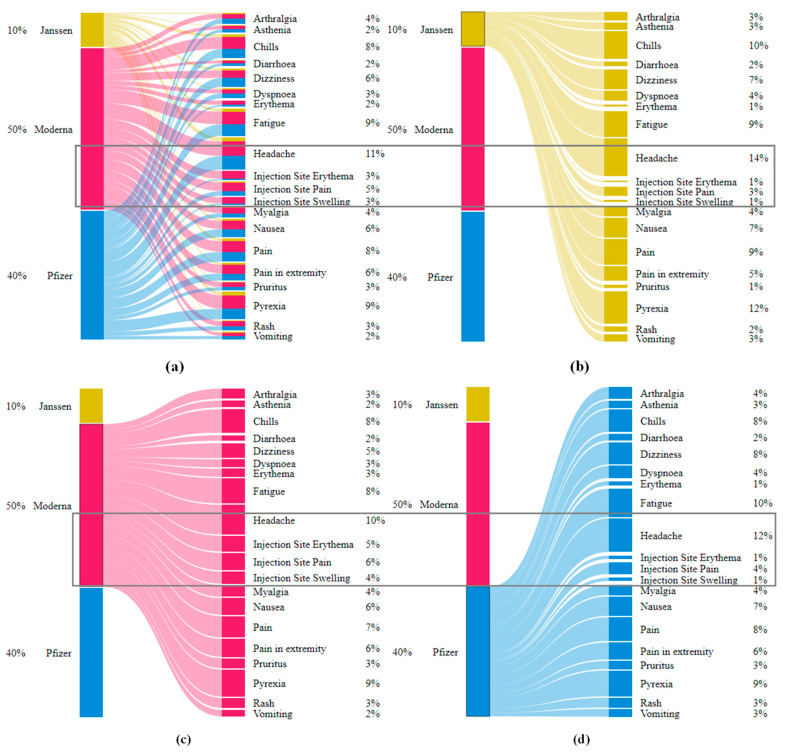
Bipartite graphs for the correlations of 20 most commonly reported AEs with the 3 vaccine producers (Pfizer-BioNTech, Moderna, and Janssen). (**a**) Shows the distribution of the entire VAERS dataset with respect to all the vaccine producers, and (**b**–**d**) show the distributions of the effects with respect to each vaccine. (**e**) Shows the correlations of 20 most commonly reported AEs with reported allergies in VAERS data. The allergies *penicillin* and *sulfa* collectively appear to be in 41% of the 643,522 VAERS reports.

**Figure 6 ijms-23-08235-f006:**
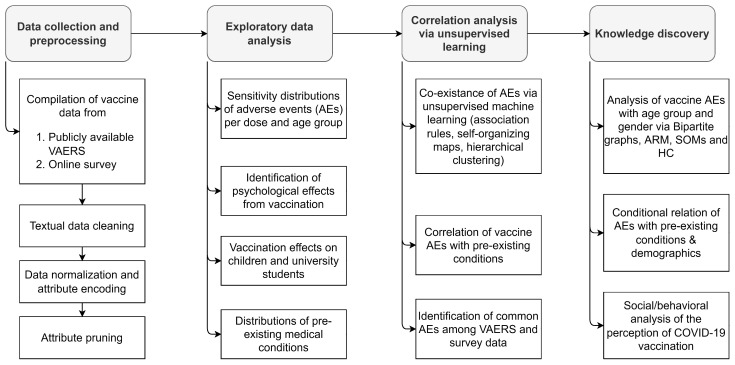
Workflow for the analysis of psychological and physical effects of COVID-19 vaccines on populations based on various demographics.

**Table 1 ijms-23-08235-t001:** Summary of the 20 most commonly reported AEs in VAERS reports and online survey data per three vaccine producers.

Effects		Vaccine Manufacturer				
VAERS	Survey Data	Pfizer-BioNTech	Pfizer-BioNTech Survey Data Points	Moderna	Moderna Survey Data Points	Janssen
Headache	Headache	48,253 (17.05%)	44 (35.77%)	51,816 (17.27%)	31 (46.97%)	15,234 (25.78%)
Pyrexia	Aches	37,418 (13.22%)	53 (43.09%)	47,476 (15,82%)	44 (66.67%)	12,811 (21.68%)
Fatigue	Tired	41,022 (14.49%)	78 (63.41%)	44,642 (14,88%)	48 (72.73%)	10,409 (17.62%)
Chills	Chills	31,965 (11.29%)	54 (43.90%)	41,313 (13.77%)	44 (66.67%)	11,001 (18.62%)
Pain	Pain in muscle	34,395 (12.15%)	49 (39.84%)	37,716 (12.57%)	34 (51.52%)	10,424 (17.64%)
Dizziness	Dizziness	32,001 (11.31%)	2 (1.63%)	25,924 (8.64%)	13 (19.70%)	8075 (13.67%)
Nausea	Nausea	28,179 (9.96%)	13 (10.57%)	29,220 (9.74%)	11 (16.67%)	7815 (13.23%)
Pain in Extremity	NA	24,708 (8.73%)	NA	31,813 (10.60%)	NA	6019 (6.11%)
Myalgia	NA	15,027 (5.31%)	NA	20,728 (6.91%)	NA	4059 (6.87%)
Arthralgia	NA	17,469 (6.17%)	NA	17,713 (5.90%)	NA	3504 (5.93%)
Injection site pain	NA	16,621 (5.87%)	NA	30,632 (10.21%)	NA	3610 (6.11%)
Dyspnoea	NA	17,612 (6.22)	NA	14,207 (4.74%)	NA	3895 (6.59%)
Rash	Itchy Skin/Rash	14,178 (5.01%)	1 (0.81%)	17,739 (5.91%)	2 (3.03%)	2194 (3.71%)
Pruritus	NA	12,013 (4.24%)	NA	17,697 (5.90%)	NA	1451 (2.46%)
Injection site erythema	NA	4685 (1.66%)	NA	27,730 (9.24%)	NA	600 (1.02%)
Asthenia	Strange Feeling	11,067 (3.91%)	10 (8.13%)	12,092 (4.03%)	15 (22.73%)	2869 (4.86%)
Vomiting	Vomiting	11,205 (3.96%)	NA	11,523 (3.84%)	2 (3.03%)	3009 (5.09%)
Injection-site swelling	Enlarged lymph nodes	4815 (1.70%)	NA	21,406 (7.14%)	1 (1.52%)	740 (1.25%)
Diarrhoea	NA	9819 (3.47%)	NA	9682 (3.23%)	NA	2018 (3.42%)
Erythema	NA	6242 (2.21%)	NA	14,227 (4.74%)	NA	838 (1.42%)

**Note:** Numbers in the table indicate the number of VAERS samples that reported corresponding AE and the percentage shows the percent of all patients in VAERS reports that were vaccinated by the given vaccine manufacturer. Survey data for Janssen were not available.

**Table 2 ijms-23-08235-t002:** The 20 most commonly reported AEs ranked with respect to the age groups and gender based on the percentage of VAERS samples reporting the corresponding AE (minimum 0% to maximum 25%). Heatmap cells are colored according to the percentage of reported samples, and the AEs are sorted according to the percentage of reported VAERS samples for age group 5–11.

Age Group (Male Participants)								Age Group (Female Participants)									
	5–11	12–15	16–18	19–30	31–50	51–65	66+		5–11	12–15	16–18	19–30	31–50	51–65	66+	25	
Pyrexia								Vomiting									
Vomiting								Headache									
Nausea								Dizziness									
Dizziness								Pyrexia									
Headache								Fatigue									
Fatigue								Injection site pain									
Injection site pain								Nausea									
Pain in Extremity								Pain in Extremity									
Rash								Rash									
Pain								Pruritus									
Chills								Pain									
Diarrhoea								Dyspnoea									
Arthralgia								Chills									
Dyspnoea								Injection site erythema									
Asthenia								Diarrhoea									
Pruritus								Asthenia									
Myalgia								Arthralgia									
Injection site erythema								Injection site swelling									
Erythema								Myalgia									
Injection site swelling								Erythema									
																	0

**Table 3 ijms-23-08235-t003:** Non-redundant association rules for post-COVID-19 vaccine AEs reported in VAERS reports for children based on cleaned with duplicate rows merged. Rule 14 was the only non-redundant many-to-one rule identified for children. The highlighted regions in gray represent a subset of rules with relatively high counts in the dataset (>200) and include {*dizziness*, *hyperhidrosis*, *syncope*, *unresponsive to stimuli*, *pyrexia*, *chills*, *myocarditis*}, which were also among the 20 most commonly reported AEs in children when explored based on their individual frequencies.

Rule	Antecedent	Consequent	Support	Confidence	Lift	Count
R-1	Flushing	Hyperhidrosis	0.016	0.80	15.18	149
R-2	Flushing	Dizziness	0.013	0.69	4.34	128
R-3	Electrocardiogram ST segment elevation	Chest pain	0.012	0.92	10.18	114
R-4	Unresponsive to stimuli	Syncope	0.023	0.72	6.35	223
R-5	Chest X-ray normal	Chest pain	0.011	0.79	8.70	103
R-6	Echocardiogram normal	Troponin increased	0.011	0.55	16.59	108
R-7	Echocardiogram normal	Chest pain	0.018	0.87	9.62	172
R-8	Myocarditis	Chest pain	0.021	0.74	8.22	205
R-9	Electrocardiogram normal	Chest pain	0.017	0.68	7.52	163
R-10	Hyperhidrosis	Dizziness	0.028	0.52	3.31	266
R-11	C-reactive protein increased	Chest pain	0.012	0.67	7.46	118
R-12	Chills	Pyrexia	0.027	0.62	5.76	263
R-13	Troponin increased	Chest pain	0.028	0.85	9.46	270
R-14	Headache, Pain	Pyrexia	0.011	0.54	4.98	101

**Table 4 ijms-23-08235-t004:** Non-redundant association rules for AEs reported in VAERS reports for the three vaccines.

Non-redundant association rules for post-COVID-19 vaccine AEs reported in VAERS reports for Pfizer-BioNTech vaccine. Rules 4–16 were non-redundant many-to-one rules identified for Pfizer-BioNTech. The highlighted regions in gray represent the subset of rules with relatively high counts in the dataset (>6000). The rules below include {*pyrexia*, *fatigue*, *headache*, *nausea*, *vomiting*, *chills*, *pain*, *myalgia*}, which were also among the 20 most commonly reported AEs for VAERS reports for Pfizer-BioNTech when explored based on their individual frequencies.						
Rule	Antecedent	Consequent	Support	Confidence	Lift	Count
R-1	Body temperature	Pyrexia	0.016	0.86	6.53	4572
R-2	Vomiting	Nausea	0.022	0.54	5.46	6095
R-3	Chills	Pyrexia	0.063	0.56	4.24	17,925
R-4	Chills, Myalgia	Fatigue	0.011	0.53	3.65	3085
R-5	Chills, Myalgia	Headache	0.013	0.62	3.61	3588
R-6	Myalgia, Pyrexia	Headache	0.013	0.58	3.40	3589
R-7	Nausea, Pain	Chills	0.012	0.50	4.46	3352
R-8	Chills, Nausea	Headache	0.018	0.60	3.52	5110
R-9	Nausea, Pain	Pyrexia	0.012	0.51	3.83	3371
R-10	Nausea, Pain	Headache	0.014	0.60	3.49	3967
R-11	Nausea, Pyrexia	Headache	0.017	0.59	3.45	4838
R-12	Fatigue, Nausea	Headache	0.019	0.57	3.36	5334
R-13	Chills, Pain	Headache	0.024	0.54	3.17	6879
R-14	Chills, Fatigue	Headache	0.025	0.56	3.26	7058
R-15	Fatigue, Pain	Headache	0.022	0.52	3.05	6130
R-16	Fatigue, Pyrexia	Headache	0.025	0.52	3.05	7008
Non-redundant association rules for post-COVID-19 vaccine AEs for Moderna vaccine. Rules 7–25 were many-to-one rules. The highlighted regions represent rules with relatively high count (>10,000). The rules below include {*pyrexia*, *headache*, *nausea*, *vomiting*, *fatigue*, *chills*, *pain*, *injection site pain*/*swelling*/*warmth*/*pruritus*/*erythema*, *myalgia*}, which were also among the 20 most commonly reported AEs for VAERS reports for Moderna when explored based on their individual frequencies.						
R-1	Injection site induration	Injection site erythema	0.01	0.66	7.19	3716
R-2	Injection site warmth	Injection site erythema	0.03	0.70	7.55	10,203
R-3	Vomiting	Nausea	0.02	0.56	5.72	6423
R-4	Injection site pruritus	Injection site erythema	0.04	0.67	7.25	13,393
R-5	Injection site swelling	Injection site erythema	0.05	0.63	6.87	13,591
R-6	Chills	Pyrexia	0.08	0.57	3.63	23,724
R-7	Injection site pruritus, Injection site warmth	Injection site swelling	0.01	0.51	7.09	3430
R-8	Injection site pain, Injection site pruritus	Injection site swelling	0.01	0.52	7.24	3177
R-9	Arthralgia, Chills	Headache	0.01	0.60	3.45	3215
R-10	Arthralgia, Fatigue	Headache	0.01	0.55	3.21	3372
R-11	Arthralgia, Pyrexia	Headache	0.01	0.57	3.27	3262
R-12	Chills, Myalgia	Headache	0.02	0.57	3.28	4685
R-13	Fatigue, Myalgia	Headache	0.02	0.55	3.20	4555
R-14	Myalgia, Pyrexia	Headache	0.02	0.53	3.07	4696
R-15	Chills, Injection site pain	Headache	0.01	0.54	3.14	3263
R-16	Chills, Pain in extremity	Headache	0.01	0.51	2.96	3515
R-17	Nausea, Pain	Chills	0.01	0.55	3.99	4193
R-18	Nausea, Pain	Pyrexia	0.01	0.55	3.47	4187
R-19	Nausea, Pain	Headache	0.02	0.60	3.50	4606
R-20	Chills, Nausea	Headache	0.02	0.60	3.48	6599
R-21	Fatigue, Nausea	Headache	0.02	0.58	3.38	6077
R-22	Nausea, Pyrexia	Headache	0.02	0.58	3.36	6198
R-23	Fatigue, Pain	Headache	0.02	0.52	3.04	6744
R-24	Chills, Fatigue	Headache	0.03	0.55	3.17	8765
R-25	Fatigue, Pyrexia	Headache	0.03	0.51	2.95	8442
Non-redundant association rules for post-COVID-19 vaccine AEs for Janssen vaccine. Rules 11–14 were many-to-one rules. The highlighted regions represent rules with relatively high count (>4000). The rules below include {*pyrexia*, *fatigue*, *headache*, *pain*, *nausea*, *chills*, *vomiting*, *myalgia*}, which were also among the 20 most commonly reported AEs for VAERS reports for Janssen when explored based on their individual frequencies.						
R-1	Body temperature	Pyrexia	0.02	0.85	3.94	1223
R-2	Decreased appetite	Fatigue	0.01	0.53	2.99	609
R-3	Vomiting	Nausea	0.03	0.54	4.12	1638
R-4	Myalgia	Headache	0.04	0.56	2.17	2268
R-5	Nausea	Headache	0.07	0.52	2.01	4051
R-6	Pain	Pyrexia	0.09	0.50	2.32	5237
R-7	Pain	Headache	0.09	0.50	1.96	5261
R-8	Chills	Headache	0.10	0.55	2.14	6058
R-9	Fatigue	Headache	0.09	0.51	1.97	5275
R-10	Pyrexia	Headache	0.11	0.51	1.99	6574
R-11	Myalgia, Nausea	Pyrexia	0.01	0.57	2.64	614
R-12	Fatigue, Myalgia	Pyrexia	0.02	0.52	2.42	942
R-13	Nausea, Pain	Chills	0.02	0.55	2.98	1318
R-14	Fatigue, Pain	Chills	0.03	0.52	2.77	1869

**Table 5 ijms-23-08235-t005:** Number of VAERS reports categorized with respect to the age group and gender along with their percentage per the three vaccine producers. For robust statistical analysis of vaccine data, duplicate reports were merged into distinct rows resulting into 643,522 rows compared to the total 905,976 reports with duplicates.

	Vaccine					
	Pfizer-BioNTech		Moderna		Janssen	
Age Group (Years)	Male	Female	Male	Female	Male	Female
5–11	233 (0.30%)	251 (0.14%)	13 (0.02%)	13 (0.01%)	3 (0.02%)	3 (0.01%)
12–15	4133 (5.36%)	4525 (2.55%)	96 (0.13%)	106 (0.05%)	43 (0.23%)	38 (0.13%)
16–18	3498 (4.54%)	4396 (2.48%)	2645 (3.59%)	3587 (1.83%)	761 (4.13%)	753 (2.52%)
19–30	10,556 (13.69%)	24,146 (13.62%)	8533 (11.58%)	21,883 (11.19%)	4054 (22.00%)	5087 (17.03%)
31–50	22,658 (29.39%)	66,760 (37.65%)	19,198 (25.06%)	64,711 (33.08%)	6561 (35.61%)	11,901 (39.83%)
51–65	18,109 (23.49%)	45,575 (25.71%)	18,917 (25.68%)	52,060 (26.62%)	4960 (26.92%)	8754 (29.30%)
66+	17,898 (23.22%)	31,645 (17.85%)	24,260 (32.93%)	53,212 (27.21%)	2045 (11.10%)	3342 (11.19%)
Total	77,085	177,298	73,662	195,572	18,427	29,878

**Note:** The number of samples per vaccine manufacturer and their percentages were calculated using clean data by removing those samples where any of the four attributes {age, gender, vaccine manufacturer, and symptom} were listed as “unknown.” There were 63,189 reports with missing age values, which were also removed from the above analysis, followed by the merger of duplicate rows in the dataset.

## Data Availability

Vaccine data from both datasets (VAERS—905,976 VAERS reports, and 211 data samples from online survey) in Excel file format are available in the submitted Appendix A. Online survey used to collect user data in PDF format is available in the submitted Appendix A (Psychological impacts and perception of vaccination and pandemic across the globe.pdf).

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
