# Peer review of "Usefulness of Vaccine Adverse Event Reporting System for Machine-Learning Based Vaccine Research: A Case Study for COVID-19 Vaccines"

_ijms, 2022, doi:10.3390/ijms23158235_

Round 1

Reviewer 1 Report

The authors utilized ML schemes including ARM and SOM to study the AE effect of three widely used covid vaccines, Pfizer-BioNTech, Moderna, and Janssen. The authors tried to identify the difference between AE of different population groups.

Some issue needs to be addressed

  1. The authors should be careful when interpreting the data. First of all, the difference could be due to sampling bais, the authors have not performed a statistical stratification in the analysis. In addition, as the authors have identified multiple factors contribute to the AE frequency when doing other comparisons, such factors should be taken into account. 

  2. The majority of the conclusion could be drawn without the ML framework.  

  3. The introduction is too long, unrelated information should be cut, and the meaning of this analysis is not clearly emphasized.

In addition, some minor points

  1. Abbreviations. For example, the term VAERS has never been introduced. The abstract section should avoid abbreviations.

  2. The author tried to describe the AE frequency difference of different vaccines, but no statistical tests were performed (Line 174-184).

  3. In the results section, some of the technical details should be only included in the methods to increase the readability.

Author Response

We thank you for your valuable review of our manuscript and input to improve our manuscript. We have addressed each comment by each reviewer in our manuscript and provided a detailed response to each of the comments in the attached Response document. Please see attached the response for more details.

We appreciate your timely review and giving us the opportunity to publish in IJMS.

Best regards,

Bilal M. Khan, PhD   Assistant ProfessorSchool of Computer Science and Engineering (JB-340)  California State University, San Bernardino 5500 University Parkway San Bernardino, CA 92407-2397 Email: Bilal.Khan@csusb.edu

Reviewer 2 Report

Dear authors, the comments are a critical review of the presented paper and should not be seen as directed to you or to your hard work.

1.       Various statistical and ML methodologies are used without being accompanied by the respective bibliographic references.

2.       The structure of the article goes directly from the introduction to the results without a theoretical framework. It is suggested a change to the structure that includes the framework of these methodologies and, eventually, mention some contexts in which they are used.

3.       It is not clear which statistical/ML software was used. It seems fair that, in case open-source software was used, the references should be included in the bibliography.

4.       The title of the article does not seem appropriate without mentioning that it is about recommendations or best practices.

Author Response

(The authors gave the same response as above.)

Round 2

Reviewer 1 Report

The authors addressed all my comments.